# Latent Instructions as Context Surrogates: Enhancing Frozen Time Series Forecasters with Instance-Adaptive Prompts

Zehao Xiao [1]  Shifeng Xie [1]  Lei Zan [1]  Malik Tiomoko [1]  Jianfeng Zhang [1]  Lujia Pan [1]
Keli Zhang [1]  Ievgen Redko [1]

## Abstract

Time series foundation models (TSFMs) benefit from longer context windows, but with steeply diminishing returns. We propose *instance-adaptive latent instructions*, a set of learnable prompt tokens prepended to the input of a frozen TSFM that serve as compact *context surrogates* to encode distributional properties. A lightweight prompt generation module, consisting of an MLP and a learnable latent token basis, maps the statistics of input series to the data-specific prompts for each instance. Introducing only ∼0.4% additional parameters and no modification to the pretrained model, our method with a short context consistently matches or outperforms the zero-shot baseline with doubled context length, while significantly reducing computational cost. Experiments on 62 long-context test sets from the GIFT-Eval benchmark with different backbones validate the effectiveness and efficiency of our approach.

## 1. Introduction

Foundation models (FMs) have advanced deep learning by providing robust pretrained architectures with strong generalization ability (Bommasani et al., 2021). Inspired by their success in language (Achiam et al., 2023; Team et al., 2023; Liu et al., 2024a) and vision (Dosovitskiy et al., 2020; Radford et al., 2021), the time series community has developed Time Series Foundation Models (TSFMs) (Woo et al., 2024; Das et al., 2024; Ansari et al., 2025) to tackle the unique challenges of large-scale temporal data.

Unlike text data, time series inherently have more dense information, leading to substantially higher computational costs under the Transformer architecture (Ansari et al.,

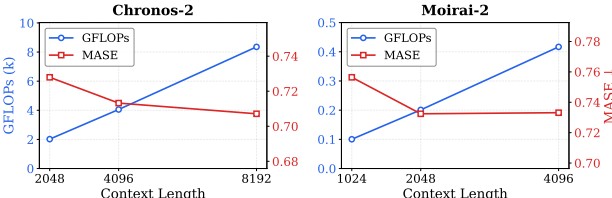

*Figure 1.* **Context scaling in TSFMs yields diminishing returns.** Computational cost (GFLOPs, blue) grows near-linearly with context length, while forecasting performance (MASE, red) improves marginally. Chronos-2 shows diminishing gains beyond context 4,096. Moirai-2's performance at 4,096 is even worse than 2,048.

2024), particularly with long contexts. Recent TSFMs adopt patch-based tokenization (Nie et al., 2023; Ansari et al., 2025; Woo et al., 2024), which groups time steps into patches, partially reducing the input sequence length and improving efficiency. Despite this, the cost of self-attention remains significant for long-context time series. Moreover, performance gains from longer contexts exhibit *diminishing returns*. In 62 long-context test datasets from the GIFT-Eval benchmark (Aksu et al., 2024), the growth in computational cost far outpaces the improvement in forecast performance for Chronos-2 and Moirai-2 (Figure 1). This suggests that much of the long context is redundant and models struggle to extract useful information from the extended history.

In the language and vision domains, prompt tuning has proven effective in steering frozen foundation models, where a small set of learnable soft tokens is trained to encode task-specific information and guide model behavior without adjusting pretrained weights (Lester et al., 2021; Li & Liang, 2021; Jia et al., 2022; Zhou et al., 2022b). Recent work further shows that such soft tokens can compress lengthy prompts into compact representations, serving as efficient surrogates for verbose context (Mu et al., 2023; Chevalier et al., 2023). In this paper, we extend this paradigm to TSFMs to enhance both their efficiency and effectiveness.

We propose **instance-adaptive latent instructions**, a set of learnable prompt tokens prepended to the input time series tokens. The tokens serve as compact **context surrogates**, encoding distributional properties (trend, seasonality, noise level) that can replace the longer context window and benefit forecasting. To achieve instance adaptiveness, we design

[1]Paris Noah's Ark Lab, Huawei. Correspondence to: Zehao Xiao <zxiao4ai@gmail.com>.

*Proceedings of the 2nd ICML Workshop on Foundation Models for Structured Data*, Seoul, South Korea. 2026. Copyright 2026 by the author(s).

a prompt generation module consisting of a lightweight MLP and a learnable latent token basis. Using the instance-specific statistical information of the input data, the module generates prompts for each input time series, providing adaptive guidance for efficient and effective forecasting.

Our contributions are as follows:

1. We propose a prompt-based framework for frozen TSFMs that introduces *latent instructions* as efficient context surrogates for long-context forecasting.

2. We design a prompt generation module that adaptively composes instance-specific latent instructions for each time series.

3. We demonstrate on 62 long-context GIFT-Eval datasets that our method with shorter context consistently matches or outperforms the baseline with doubled context, while significantly reducing computational cost.

## 2. Method

**Notation.** Let $f_\theta$ denote a pretrained TSFM with frozen parameters $\theta$. Given a univariate context $\mathbf{x} = (x_1, \ldots, x_T) \in \mathbb{R}^T$, the model first applies instance normalization to obtain $\tilde{\mathbf{x}} = (\mathbf{x} - \mu)/\sigma$. The normalized context $\tilde{\mathbf{x}}$ is partitioned into $L = \lceil T/P \rceil$ patches of size $P$ and mapped to token embeddings $\mathbf{E} = (\mathbf{e}_1, \ldots, \mathbf{e}_L) \in \mathbb{R}^{L \times d}$, where $d$ is the model dimension. A Transformer encoder processes $\mathbf{E}$ and predicts quantile forecasts $\hat{\mathbf{y}} \in \mathbb{R}^{Q \times H}$ for $Q$ quantile levels over horizon $H$.

Given the diminishing returns of context scaling in TSFMs (Figure 1), we propose *instance-adaptive latent instructions*, which are learnable prompt tokens that efficiently encode distributional information to replace longer contexts. An overall framework is shown in Figure 2.

### 2.1. Prompt Injection as Latent Instructions

Inspired by soft prompting in LLMs and VLMs (Lester et al., 2021; Li & Liang, 2021; Zhou et al., 2022b), where a small set of continuous tokens can efficiently steer a frozen model as task-specific instructions, we propose to augment the input of TSFMs with $M$ learnable prompt tokens $\mathbf{P} = (\mathbf{p}_1, \ldots, \mathbf{p}_M) \in \mathbb{R}^{M \times d}$ prepended to the patch embeddings:

$$\mathbf{H} = f_\theta([\mathbf{P}; \mathbf{E}]), \qquad (1)$$

where $[\cdot; \cdot]$ denotes concatenation along the sequence dimension. During training, only the prompts $\mathbf{P}$ are optimized. All parameters $\theta$ of the pretrained TSFM remain frozen. These prompt tokens are trained to serve as **latent instructions** that encode the distributional properties, such as trend, seasonality, and noise characteristics, to guide the forecasting of the frozen model. Without modifying any model

weights, the pretrained general forecasting capabilities are fully preserved. Moreover, the latent instructions capture the essential information contained in long contexts in a much more compact form, therefore effectively replacing redundant history and improving both forecasting performance and computational efficiency.

**Position encoding.** In Transformer-based TSFMs, position encodings indicate the temporal order of the input tokens. We assign all prompt tokens position index 0, making them position-agnostic conditioning signals that are not tied to any specific temporal location. The time series tokens retain their original relative order with indices $(1, 2, \ldots, L)$.

### 2.2. Instance-Adaptive Prompt Generation

In its simplest form, $\mathbf{P}$ is a set of learnable tokens shared by all inputs. While effective for datasets with similar properties, a static prompt cannot capture the variation across individual series and is prone to overfitting. To address the problem, we further propose instance-adaptive prompt generation that constructs tailored latent instructions for each input.

Our instance-adaptive prompt consists of two components:

$$\mathbf{P}_i = \underbrace{\mathbf{P}_s}_{\text{shared}} + \underbrace{\sigma(g) \cdot \mathbf{P}_a(\mathbf{x})}_{\text{instance-adaptive}}, \qquad (2)$$

where $\mathbf{P}_s \in \mathbb{R}^{M \times d}$ is a learnable prompt that captures shared distributional information across the training data, and $\mathbf{P}_a(\mathbf{x})$ is an input-dependent refinement that tailors the prompt tokens to each individual series. $\sigma(g)$ is a sigmoid gate with a trainable parameter $g$ that controls the magnitude of the refinement.

**Adaptive prompt generation.** To generate the adaptive component $\mathbf{P}_a(\mathbf{x})$, we extract statistical features from the input series that characterize its distributional properties, such as trend, seasonality, autocorrelation patterns, and noise level. Specifically, from the normalized data $\tilde{\mathbf{x}}$, we extract a compact feature vector $\mathbf{s} \in \mathbb{R}^{N_s}$, comprising scale-invariant statistics that characterize the location, spread, shape, and temporal structure of the input (details in Appendix C). All statistics are computed non-parametrically in normalized space, ensuring invariance to the location and scale of the input series.

We then generate the adaptive prompt component $\mathbf{P}_a(\mathbf{x})$ from the statistics $\mathbf{s}$ by a lightweight MLP. Since $\mathbf{s}$ is low-dimensional, directly mapping it to the full prompt space $\mathbb{R}^{M \times d}$ requires a disproportionately large output layer. We instead factorize the generation through a **latent token basis** with $r$ learnable prompt patterns $\{\mathbf{T}_1, \ldots, \mathbf{T}_r\} \subset \mathbb{R}^{M \times d}$ and a **data-specific composition vector** $\mathbf{z} \in \mathbb{R}^r$, which assembles the adaptive component as a linear combination

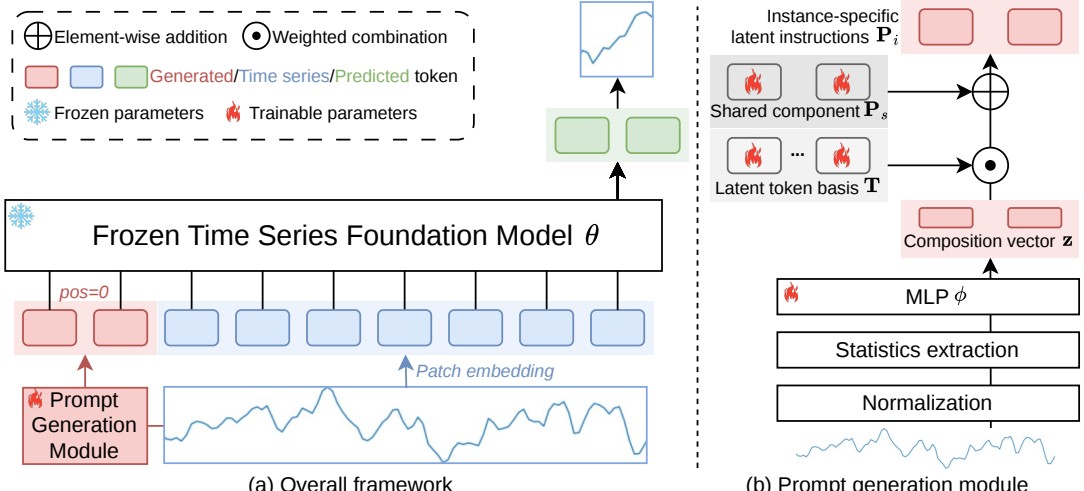

*Figure 2.* **Overall framework of instance-adaptive latent instructions.** (a) The learnable latent instructions (red) are constructed from time series via a prompt generation module and prepended at position 0 to the patch embeddings (blue) of a frozen TSFM. (b)The prompt generation module consists of the normalization and statistics extraction of the input time series, as well as a learnable lightweight MLP, latent token basis, and the shared prompt component to generate the final latent instructions.

of the basis:

$$\mathbf{P}_a(\mathbf{x}) = \sum_{i=1}^{r} z_i \cdot \mathbf{T}_i, \qquad \mathbf{z} = \text{MLP}_\phi(\mathbf{s}) \in \mathbb{R}^r, \quad (3)$$

where $\text{MLP}_\phi \colon \mathbb{R}^{N_s} \to \mathbb{R}^r$ is a two-layer network:

$$\text{MLP}_\phi(\mathbf{s}) = \mathbf{W}_2 \, \text{LN}\big(\text{GELU}(\mathbf{W}_1\mathbf{s} + \mathbf{b}_1)\big) + \mathbf{b}_2 , \quad (4)$$

with $\mathbf{W}_1 \in \mathbb{R}^{h \times N_s}$ and $\mathbf{W}_2 \in \mathbb{R}^{r \times h}$. This constrains the instance-level variation to an $r$-dimensional subspace, providing implicit regularization while keeping the number of input-dependent parameters minimal.

The composed prompt $\mathbf{P}_i$ integrates both shared knowledge through $\mathbf{P}_s$ and instance-specific information through $\mathbf{P}_a(\mathbf{x})$, enabling generalizability across series and adaptability to each input. By distilling the statistical information that is implicitly contained in long contexts, the generated prompt tokens serve as compact **context surrogates** for each input time series, reducing the computational cost of processing extended history.

### 2.3. Training

**Loss function.** Following the pretraining objective of existing TSFMs (Ansari et al., 2025; Woo et al., 2024), we minimize the quantile loss over the forecast horizon:

$$\mathcal{L} = \frac{1}{QH} \sum_{q=1}^{Q} \sum_{t=1}^{H} \rho_{\tau_q}\Big(y_t - \hat{y}_t^{(q)}\Big), \quad (5)$$

where $\rho_\tau(u)$ is the pinball loss. During training, only the parameters of the prompt generation module are optimized. The base TSFM $f_\theta(\cdot)$ remains entirely frozen.

## 3. Experiments

**Data.** We evaluate on the GIFT-Eval benchmark (Aksu et al., 2024), selecting 22 datasets with context lengths exceeding 2048, yielding 62 test sets on short, medium, and long-term forecasting horizons. Our method is trained and evaluated on the combination of these datasets. Detailed dataset information are provided in Appendix B. Training, validation, and test sets follow the standard time-based splitting of GIFT-Eval: the first 80% of each series for training, the next 10% for validation, and the final 10% for testing, preserving temporal ordering and preventing data leakage.

**Implementation details.** We optimize the prompt generation module by AdamW (Loshchilov & Hutter, 2019) with learning rate $10^{-3}$ and batch size 64, training for up to 50 epochs with early stopping on validation loss. We use $M = 100$ prompt tokens, basis size $r = 4$, hidden dimension $h = 32$.

**Baselines.** We compare our method with several baselines: (1) *Zero-shot*: the frozen TSFM; (2) *Full fine-tuning*: updating all model parameters; and (3) *LoRA* (Hu et al., 2022): low-rank adaptation of the encoder's attention layers. All baselines use the same training data and evaluation protocol. For LoRA and full fine-tuning, we report results with the best learning rate selected via validation.

**Metrics.** We report MASE (Mean Absolute Scaled Error) and CRPS (Continuous Ranked Probability Score) for forecasting performance, as well as GFLOPs for efficiency.

### 3.1. Results

**Comparison with baselines.** We compare our latent instructions with zero-shot, full fine-tuning, and LoRA on

*Table 1.* **Comparison on 62 long-context GIFT-Eval test sets using Chronos-2.** $\Delta$ denotes relative improvement over the zero-shot baseline at the same context length. Our method outperforms all baselines with different context length.

| Method | MASE ↓ | $\Delta$ | CRPS ↓ | $\Delta$ |
|---|---|---|---|---|
| *Context length=2048* | | | | |
| Zero-shot | 0.728 | – | 0.489 | – |
| Full FT | 0.717 | 1.5% | 0.473 | 3.3% |
| LoRA | 0.716 | 1.6% | 0.475 | 2.9% |
| **Ours** | **0.710**$_{\pm.001}$ | **2.5%** | **0.469**$_{\pm.001}$ | **4.1%** |
| *Context length=4096* | | | | |
| Zero-shot | 0.713 | – | 0.478 | – |
| Full FT | 0.708 | 0.7% | 0.477 | 0.2% |
| LoRA | 0.710 | 0.4% | 0.469 | 1.9% |
| **Ours** | **0.703**$_{\pm.001}$ | **1.4%** | **0.466**$_{\pm.002}$ | **2.5%** |
| *Context length=8192* | | | | |
| Zero-shot | 0.707 | – | 0.470 | – |
| Full FT | 0.704 | 0.4% | 0.476 | -1.3% |
| LoRA | 0.705 | 0.3% | 0.470 | 0.0% |
| **Ours** | **0.701**$_{\pm.001}$ | **0.8%** | **0.465**$_{\pm.001}$ | **1.1%** |

*Table 2.* **Efficiency comparison on Chronos-2.** $\Delta$ is relative to the zero-shot baseline at doubled context. Our method outperforms the zero-shot baseline with doubled contexts, while obviously reducing computational cost, up to 32% GFLOPs.

| Setting | MASE↓ | CRPS↓ | GFLOPs↓ |
|---|---|---|---|
| Zeor-shot (context len=4096) | 0.713 | 0.478 | 4053 |
| **Ours (context len=2048)** | **0.710**$_{\pm.001}$ | **0.469**$_{\pm.001}$ | 3569 |
| $\Delta$ | *0.5%* | *1.9%* | *12%* |
| Zeor-shot (context len=8192) | 0.707 | 0.470 | 8355 |
| **Ours (context len=4096)** | **0.703**$_{\pm.001}$ | **0.466**$_{\pm.002}$ | 5664 |
| $\Delta$ | *0.6%* | *0.9%* | *32%* |

Chronos-2 over three context lengths. As shown in Table 1, all adaptation methods improve MASE over the zero-shot baseline, with gains diminishing at longer contexts. full fine-tuning and LoRA yield smaller MASE improvements and unstable CRPS (even degrading in some cases). In contrast, our method consistently achieves the best MASE and CRPS at every context length, demonstrating that our latent instructions are more stable than weight-space modification methods. Notably, our method at context 2048 (MASE 0.710) already outperforms the zero-shot baseline at context 4096 (MASE 0.713), and similarly at context 4096 versus the 8192 baseline, effectively reducing the required context length.

**Efficiency analysis.** We demonstrate the efficiency of our latent instructions as replacements for longer contexts. As shown in Table 2, our method consistently outperforms the zero-shot baseline with doubled context length, while significantly reducing GFLOPs. In particular, the method with context 4096 saves 32% GFLOPs compared to the baseline with context 8192, with better MASE and CRPS. Moreover, the prompt generation module adds only ~400K parameters (0.39% of Chronos-2's 119M parameters), operating entirely in the input space without modifying any pretrained weights. Additional efficiency results are provided in Appendix D.

*Table 3.* **Results on Moirai-2** with 62 long-context GIFT-Eval test sets. Our method again outperforms the zero-shot baseline with both the same and doubled context lengths.

| Setting | MASE ↓ | $\Delta$ | CRPS ↓ | $\Delta$ |
|---|---|---|---|---|
| Zero-shot (context len=2048) | 0.732 | – | 0.503 | – |
| Zero-shot (context len=4096) | 0.733 | – | 0.504 | – |
| **Ours (context len=2048)** | **0.727** | 0.8% | **0.496** | 1.6% |

*Table 4.* **Overall and per-dataset trained latent instructions.** Per-dataset training yields stronger results on both backbones, indicating further potential with dataset-specific adaptation.

| Settings | MASE ↓ | $\Delta$ | CRPS ↓ | $\Delta$ |
|---|---|---|---|---|
| *Base model: Chronos-2 Context length=4096* | | | | |
| Zero-shot | 0.713 | – | 0.478 | – |
| Overall training | 0.703 | 1.4% | 0.466 | 2.5% |
| Per-dataset training | 0.699 | 2.0% | 0.460 | 3.8% |
| *Base model: Moirai-2 Context length=2048* | | | | |
| Zero-shot | 0.732 | – | 0.503 | – |
| Overall training | 0.727 | 0.8% | 0.496 | 1.6% |
| Per-dataset training | 0.724 | 1.1% | 0.494 | 1.8% |

**Generalization to other TSFM.** To demonstrate generalization across architectures, we also evaluate our method on Moirai-2, a decoder-only TSFM. As shown in Table 3, the zero-shot baseline at context 4096 even degrades compared to 2048, illustrating the diminishing returns of context scaling. Our method at context 2048 improves over the zero-shot baseline at both context lengths, confirming that the latent instructions generalize across TSFM architectures.

**Per-dataset training.** The previous experiments train a single shared prompt generation module across 22 datasets. Beyond this, we further evaluate per-dataset tuning, where a separate module is trained and evaluated for each dataset. As shown in Table 4, per-dataset training consistently yields stronger results on both backbones, indicating the challenges of a single shared module in generalizing across diverse datasets, and that our method has potentially stronger capability with dataset-specific adaptation. We provide more ablation studies in Appendix D.

## 4. Conclusion

We proposed *latent instructions* for pretrained time series foundation models, a prompt-based framework that enhances frozen TSFMs by prepending learnable tokens as compact context surrogates. A lightweight prompt generation module constructs instance-adaptive prompts for each input time series conditioned on its statistical properties to provide data-specific guidance and avoid overfitting. Experiments on 62 long-context GIFT-Eval test sets show that our method with shorter context consistently matches or outperforms zero-shot baselines with doubled context length, while significantly reducing computational cost. Our current work focuses on enhancing forecasting performance and efficiency for long-context series. Extending it to short-context scenarios is a promising direction for future work.

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

# Appendix

# A. Related work

**Transformer-based time series foundation models.** The success of Transformer-based architectures (Vaswani et al., 2017) in NLP has inspired a growing body of foundation models for time series (Liang et al., 2024). Early approaches propose to tokenize individual time steps for forecasting as LLMs (Rasul et al., 2023; Ansari et al., 2024; Chang et al., 2025). To reduce sequence length and capture local temporal patterns, inspired by vision transformer (Dosovitskiy et al., 2020), PatchTST (Nie et al., 2023) proposes patch-based tokenization before the Transformer encoder, which is adopted by most recent TSFMs, such as MOMENT (Goswami et al., 2024), Moirai (Woo et al., 2024), TimesFM (Das et al., 2024), Toto (Cohen et al., 2024), and Timer (Liu et al., 2024b). Recent work continues to scale these models using MoE (Liu et al., 2026) or hierarchical architectures (Sun et al., 2025). Recently, beyond the common decoder-only architecture like Moirai-2 and TimesFM, there are also encoder-only emerging such as Chronos-2 (Ansari et al., 2025) and PatchTST-FM (Wen et al., 2026). Despite patching, extending context length remains computationally expensive with diminishing returns, motivating efficient alternatives for leveraging long historical information.

**Parameter-efficient fine-tuning.** Adapting large pretrained models without full fine-tuning has been extensively studied in NLP and vision tasks. Prompt tuning (Lester et al., 2021) and prefix-tuning (Li & Liang, 2021) prepend learnable soft tokens to guide frozen language models. LoRA (Hu et al., 2022) injects low-rank updates into attention weights as an alternative. In vision, visual prompt tuning (Jia et al., 2022) extends soft prompts to vision Transformers. CoOp (Zhou et al., 2022b) and CoCoOp (Zhou et al., 2022a) learn prompts for vision-language models. Recently, in time series, Gen-P-Tuning (Liu et al., 2025) applies static prompt tuning to adapt univariate TSFMs for multivariate healthcare data. TimesFM-ICF (Faw et al., 2025) learns to use in-context examples to adapt pretrained forecasters. Our work differs from these approaches in two key aspects: our latent instructions are *instance-adaptive* rather than static, and our goal is efficient context scaling rather than domain or cross-variate adaptation.

# B. Dataset Details

We summarizes the 22 training datasets and corresponding 62 test sets used in our experiments in Table 5, all sourced from the GIFT-Eval benchmark (Aksu et al., 2024) with context lengths exceeding 2048. Each dataset is evaluated on up to three forecasting horizons (short, medium, long), depending on the available context length.

*Table 5.* **Overview of training datasets and test sets.** All of the datasets have context longer than 2048. Each training dataset yields 1–3 test sets depending on available forecasting horizons.

| Training Dataset | Freq. | Test Sets (horizons) |
|---|---|---|
| bitbrains_fast_storage | 5T | short, medium, long |
| bitbrains_rnd | 5T | short, medium, long |
| bizitobs_application | 10S | short, medium, long |
| bizitobs_l2c | 5T | short, medium, long |
| bizitobs_service | 10S | short, medium, long |
| electricity | 15T | short, medium, long |
| electricity | H | short, medium, long |
| ett1 | 15T | short, medium, long |
| ett1 | H | short, medium, long |
| ett2 | 15T | short, medium, long |
| ett2 | H | short, medium, long |
| jena_weather | 10T | short, medium, long |
| jena_weather | H | short, medium, long |
| kdd_cup_2018 | H | short, medium, long |
| loop_seattle | 5T | short, medium, long |
| loop_seattle | H | short, medium, long |
| m_dense | H | short, medium, long |
| saugeen | D | short |
| solar | 10T | short, medium, long |
| solar | H | short, medium, long |
| sz_taxi | 15T | short, medium, long |
| us_births | D | short |
| **Total: 22 datasets** | | **62 test cases** |

*Table 6.* **Statistics used in the prompt generation module.** All computed from the normalized context $\tilde{x}$.

| Index | Statistics | Description |
|---|---|---|
| 0 | Mean | Mean of normalized values |
| 1 | Std | Standard deviation |
| 2 | Min | Minimum value |
| 3 | Max | Maximum value |
| 4 | Trend | Difference between second-half and first-half means |
| 5 | Diff mean | Mean of first-order differences |
| 6 | Diff std | Std of first-order differences |
| 7 | Autocorr (lag-1) | Lag-1 autocorrelation |
| 8 | Frac. observed | Fraction of non-missing values |
| 9 | Log length | Log of context length (normalized) |

## C. Statistics Features

In Table 6 we list the $N_s = 10$ scale-invariant statistics extracted from the normalized context $\tilde{x}$ to form the feature vector $s$. All statistics are computed non-parametrically and are invariant to the location and scale of the original series.

## D. Additional experiments

### D.1. Ablation Studies

**Ablation on instance-adaptive prompts.** To demonstrate the effectiveness of our instance-adaptive prompts, we com-

*Table 7.* **Ablation on instance-adaptive prompts** on Chronos-2 with context length 4096. Instance-adaptive prompts perform better than only the shared ones.

| Method | MASE ↓ | CRPS ↓ |
|---|---|---|
| Zero-shot | 0.713 | 0.478 |
| Shared prompt $\mathbf{P}_s$ | 0.708 | 0.471 |
| **Instance-adaptive $\mathbf{P}_i$** | $\mathbf{0.703}_{\pm.001}$ | $\mathbf{0.466}_{\pm.002}$ |

*Table 8.* **Ablation on number of prompt tokens $M$.** Experiments are conducted on Chronos-2 with context 4096.

| Num. of prompt tokens $M$ | MASE ↓ | CRPS ↓ |
|---|---|---|
| 50 | $0.705_{\pm.001}$ | $0.467_{\pm.003}$ |
| **100** | $\mathbf{0.703}_{\pm.001}$ | $\mathbf{0.466}_{\pm.002}$ |
| 200 | $0.706_{\pm.001}$ | $0.472_{\pm.003}$ |

pare against using only the shared prompt $\mathbf{P}_s$ on Chronos-2 with context 4096. As shown in Table 7, both variants outperform the zero-shot baseline, while the instance-adaptive prompts further improve over the shared ones. This shows that the instance-specific component provides meaningful per-instance guidance beyond shared information.

**Ablation on prompt length.** We also conduct ablation studies on the number of prompt tokens $M$. Table 8 shows the performance with $M = 50/100/200$ on Chronos-2 with context 4096. $M = 100$ achieves the best MASE, while $M = 50$ is competitive with slightly worse MASE and CRPS. $M = 200$ performs worse on both MASE and CRPS, indicating that 100 tokens provide sufficient capacity to encode the relevant distributional information. Additional tokens can lead to overfitting due to the increased parameter count.

**Extra efficiency analysis.** Beyond GFLOPs, we also evaluate time and memory costs under the same setting as Table 2. As shown in Table 9, our method achieves better performance with lower costs compared to the zero-shot baseline with doubled context length. In particular, with context 4096 our method saves over 20% in latency and GPU memory and over 30% in GFLOPs compared to the baseline with context 8192, while achieving better forecasting accuracy.

*Table 9.* **Further efficiency comparison on Chronos-2.** Our method outperforms the baseline with doubled context, while obviously reducing computational cost.

| Setting | MASE↓ | CRPS↓ | Time | Mem | GFLOPs |
|---|---|---|---|---|---|
| Zero-shot (context length=4096) | 0.713 | 0.478 | 106.9 ms | 1142 MB | 4053 |
| **Ours (context length=2048)** | **0.710**$_{\pm.001}$ | **0.469**$_{\pm.001}$ | 102.4 ms | 1108 MB | 3569 |
| Δ | *0.5%* | *1.9%* | *4%* | *3%* | *12%* |
| Zero-shot (context length=8192) | 0.707 | 0.470 | 218.2 ms | 1783 MB | 8355 |
| **Ours (context length=4096)** | **0.703**$_{\pm.001}$ | **0.466**$_{\pm.002}$ | 160.8 ms | 1429 MB | 5664 |
| Δ | *0.6%* | *0.9%* | *26%* | *20%* | *32%* |

