# OpenReview forum: "Latent Instructions as Context Surrogates: Enhancing Frozen Time Series Forecasters with Instance-Adaptive Prompts"
_ICML.cc/2026/Workshop/FMSD — FMSD @ ICML 2026 SpotlightOral_

### Official Review · Reviewer_TvUX · 2026-05-13
**Review for "Latent Instructions as Context Surrogates: Enhancing Frozen Time Series Forecasters with Instance-Adaptive Prompts"**

**Rating:** 6
**Confidence:** 4

**Review:**

### **Summary**

This paper investigates the “scaling trap” in Time Series Foundation Models (TSFMs), where extending the context window yields only limited forecasting improvements while substantially increasing computational cost. To address this issue, the authors introduce **instance-adaptive latent instructions**, a lightweight prompting mechanism that prepends learnable tokens as compact representations of long-range context.

The proposed prompt generation module extracts scale-invariant statistics from the input and dynamically constructs latent instructions using an MLP and a latent token basis. Experiments on the GIFT-Eval benchmark demonstrate that shorter-context models equipped with the proposed method can match or outperform zero-shot baselines using twice the context length, while also reducing GFLOPs and memory consumption.

------

### **Strengths**

- **The paper is well written.**
- **The paper addresses an important and timely challenge**: the inefficiency of long-context scaling in TSFMs.
- The proposed approach provides notable efficiency improvements, achieving up to **32% lower GFLOPs** and **20% less GPU memory usage** while maintaining competitive forecasting performance.
- The method is highly parameter-efficient, requiring only about **0.4%** additional parameters while keeping the backbone model frozen.

------

### **Weaknesses**

- Although the paper is framed as addressing diminishing returns in long-context scaling, **the results suggest that the method may instead reach saturation earlier.** For instance, in Chronos-2, the zero-shot baseline continues to improve more consistently as the context length increases compared to the proposed approach.
- **The experiments do not include extremely large context windows (e.g., 16k+)**, making it difficult to determine whether the method fundamentally improves scaling behavior or merely accelerates short-term saturation.
- **Comparisons with recent context compression or soft-prompting approaches for time-series models are limited**, which weakens the evaluation of the method’s novelty and relative advantage.

------

### **Justification of Score**

The paper is motivated by a meaningful and practically relevant problem and demonstrates clear computational benefits. However, the current experimental results do not fully support the claim that the method fundamentally resolves the diminishing-returns issue in long-context scaling.

Instead, the approach appears closer to an effective and efficient **context-substitution strategy** rather than a direct remedy for scaling limitations. Additional evaluations at substantially larger context lengths would strengthen the paper’s central claim.

---

### Official Review · Reviewer_S4B6 · 2026-05-18

**Rating:** 7
**Confidence:** 4

**Review:**

## Summary
The paper proposes a technique to improve the performance of time series foundation models in long-context settings. The approach adds a learnable prefix to the input, where statistics of the input time series are mapped through an MLP into prompt tokens. The module can be trained on top of existing TSFMs without modifying the pretrained weights. Experiments on GIFT-Eval show that the proposed approach improves forecasting accuracy compared to full fine-tuning and LoRA-based approaches.

## Strengths
The idea is interesting and well-motivated. The experimental design is solid and supports the main claims of the paper through comparisons with multiple fine-tuning strategies. The paper is also well written and easy to follow.

## Areas for Improvement
I did not identify any major flaws in the paper, but several aspects could be clarified to strengthen the empirical evaluation and better demonstrate the practical benefits of the proposed approach (see detailed comments).

## Detailed Comments
1. All approaches are evaluated only using the geometric mean of relative errors for MASE and CRPS. While the proposed method shows overall improvements, it is unclear whether these gains come from large improvements on a small subset of datasets or from consistent improvements across most tasks. To better characterize the results, I recommend including an additional statistic such as pairwise win rates between methods.
2. Table 1 appears to compare fine-tuned model variants trained jointly across all datasets. What happens if the baseline fine-tuning methods are trained separately per dataset? Tables 3 and 4 analyze the proposed approach in the per-dataset setting, but it would also be useful to include the corresponding fine-tuning baselines for a fair comparison.
3. One of the main advantages of pretrained models is reducing the need for task-specific training. It would therefore be interesting to evaluate how well the proposed approach generalizes across tasks and datasets. For example, the authors could train the prompt-generation module on a subset of GIFT-Eval datasets and evaluate on held-out datasets representing unseen task types.

## Justification of Score
The paper proposes an interesting idea that is clearly relevant to the workshop theme. The methodology appears sound, and the main conclusions are supported by the experimental results.

---

### Official Review · Reviewer_PmU7 · 2026-05-20
**Latent Instructions as Context Surrogates: Enhancing Frozen Time Series Forecasters with Instance-Adaptive Prompts**

**Rating:** 10
**Confidence:** 4

**Review:**

TSFMs become increasingly computationally expensive, yet forecast accuracy improves with diminishing returns as the context window grows. The authors propose a method to achieve higher forecast accuracy not by increasing the context window, but by providing a short prompt that summarizes time series statistics. This prompt is trained on the dataset while the TSFM weights remain frozen. The method was evaluated on Chronos-2 and Moirai-2 TSFMs using the GIFT-Eval dataset, demonstrating that forecast accuracy improves without increasing the context window.

The approach is highly interesting, can be readily utilized by practitioners to improve forecast accuracy, and may inspire novel methods for enhancing TSFMs and other foundation models.